# Non-Invasive Assessment of Skin Surface Proteins of Psoriasis Vulgaris Patients in Response to Biological Therapy

**DOI:** 10.3390/ijms242216248

**Published:** 2023-11-13

**Authors:** Kadri Orro, Kristiina Salk, Anna Merkulova, Kristi Abram, Maire Karelson, Tanel Traks, Toomas Neuman, Pieter Spee, Külli Kingo

**Affiliations:** 1Department of Chemistry and Biotechnology, Tallinn University of Technology, Akadeemia tee, 12618 Tallinn, Estonia; kadrikene@gmail.com; 2FibroTx LLC., Mäealuse 4, 12618 Tallinn, Estoniaanna.biotrix@gmail.com (A.M.); pieter.spee@ps-pharmaconsult.com (P.S.); 3Clinic of Dermatology, Tartu University Hospital, 50417 Tartu, Estonia; 4Clinic of Dermatology, Institute of Clinical Medicine, Tartu University, 50417 Tartu, Estonia; 5PS! Pharmaconsult, Moellemoseparken 44, 3450 Alleroed, Denmark

**Keywords:** biomarker, transdermal analysis patch, psoriasis, inflammation, dermatology, treatment monitoring, biological therapy

## Abstract

Measurements of skin surface biomarkers have enormous value for the detailed assessment of skin conditions, both for clinical application and in skin care. The main goals of the current study were to assess whether expression patterns of skin surface hBD-1, hBD-2, IL-1α, CXCL-1, and CXCL-8, examples of proteins known to be involved in psoriasis pathology, are associated with disease severity and whether expression patterns of these proteins on the skin surface can be used to measure pharmacodynamic effects of biological therapy. In this observational study using transdermal analysis patch (TAP), levels of skin surface IL-1α, hBD-1, hBD-2, CXCL-1/2, and CXCL-8 of psoriasis vulgaris (PV) patients over biological therapy were assessed. The Psoriasis Area Severity Index (PASI) and local score for erythema, induration, and desquamation were determined from the exact same skin area as FibroTx TAP measurements. Thirty-seven adult PV patients were included, of which twenty-three were subjected to anti-TNF-α, seven to anti-IL-17A, and seven to anti-IL12/IL-23 therapy. Significantly higher levels of hBD-1, hBD-2, CXCL-1/2, and CXCL-8 were detected on lesional skin compared to the non-lesional skin of the PV patients. In contrast, lower levels of IL-1α were found in lesional skin compared to non-lesional skin. In addition, we observed that the biomarker expression levels correlate with disease severity. Further, we confirmed that changes in the expression levels of skin surface biomarkers during biological therapy correlate with treatment response. Biomarker expression patterns in response to treatment differed somewhat between treatment subtypes. We observed that, in the case of anti-TNF-α therapy, an increase after a steady decrease in the expression levels of CXCL-1/2 and CXCL-8 occurred before the change in clinical scores. Moreover, response kinetics of skin surface proteins differs between the applied therapies—hBD2 expression responds quickly to anti-IL-17A therapy, CXCL-1/2 to anti-IL-12/23, and levels of CXCL-8 are rapidly down-regulated by IL-17A and IL-12/23 therapy. Our findings confirm that the skin surface hBD-2, IL-1α, CXCL-1/2, and CXCL-8 are markers for the psoriasis severity. Further, data obtained during this study give the basis for the conclusion that skin surface proteins CXCL-1/2 and CXCL-8 may have value as therapeutic biomarkers, thus confirming that measuring the ‘molecular root’ of inflammation appears to have value in scoring disease severity on its own.

## 1. Introduction

Psoriasis is a common but complex chronic inflammatory skin disorder affecting up to an estimated 3% of the world’s population (Boehncke and Schön, 2015) [1]. The most prevalent form of psoriasis, called *psoriasis vulgaris* (PV) or plaque psoriasis, is characterized by the manifestation of well-defined erythematous plaques covered with white scales. Moreover, psoriasis is often associated with comorbidities like psoriatic arthritis, metabolic syndrome, and cardiovascular disease (Gelfand et al., 2007 [2]; Takeshita et al., 2017 [3,4]), and the visual appearance of psoriatic lesions can severely impact the patient’s psychological state and quality of life (Boehncke and Schön, 2015) [1].

The molecular mechanism of the disease covers defective regulation of epidermal keratinocytes leading to their hyperproliferation, altered protein expression, loss of a mature granular layer, and parakeratosis. In addition, infiltrating neutrophils, inflammatory dendritic cells, and effector T-cells Th1, Th17, and Th22 further drive the inflammation and contribute to clinical symptoms, causing a permanent state of chronic inflammation in affected skin (Schön and Boehncke, 2005) [5].

The type of applied psoriasis treatment depends on the severity of the disease. Therefore, therapy options range from mild topical therapy to more harsh therapies for moderate and severe psoriasis, such as phototherapy (e.g., ultraviolet B (UVB)), photochemotherapy (e.g., psoralen ultraviolet A (PUVA)), systemic treatments with conventional agents (e.g., methotrexate, cyclosporine, and retinoids), and biological treatments (e.g., anti-tumor necrosis factor (anti–TNF-α), interleukin inhibitors (anti-IL17, anti-IL-12/23, and anti-IL23)), (Patel et al., 2009 [6], Kim et al., 2012 [7], Laws et al., 2012 [8]; Rustin et al., 2012 [9], Kamata et al., 2020 [10]. 

The clinical evaluation of the efficacy of treatment is primarily performed visually using the Psoriasis Area Severity Index (PASI). For years, the PASI has been considered to be the gold standard to quantify disease severity, taking into account the affected body surface area along with scaling, redness, and cellular infiltration. Histological evaluation is conducted in borderline cases (Griffiths and Barker, 2007 [11]). However, such an approach has its limitations, like limited coverage and insensitivity to small changes, for example (Clinical Review Report, 2017) [12]. Therapeutic efficacy is defined as a reduction in psoriasis clinical scores; it does not occur instantly, it is reflected on the skin surface after a while, and it does not reflect the molecular state of the underlying inflammation in the deeper layers of the skin. 

To date, there is relatively little information regarding the skin surface protein expression patterns on the skin surface found on psoriatic lesions during therapy and whether the expression of such skin surface proteins can be used to measure the pharmacodynamic effects of biological therapy. Human skin surface contains a variety of signaling molecules, such as interleukins, chemokines, cytokines, and antimicrobial peptides, that drive the biological processes underlying the visual hallmarks of skin diseases (Jansen et al., 2009 [13]; Baliwag et al., 2015 [14]; Mehul et al., 2017 [15]; Chularojanamontri et al., 2019 [16]). As such, these “molecular footprints” could be suitable for the development of diagnostic methods that can monitor disease progression and measure the response to treatment over the course of therapy directly at the site of inflammation.

We have previously introduced a non-invasive method, the FibroTx Transdermal Application Patch (TAP), to measure soluble biomarkers directly from the skin surface. TAP consists of an adhesive bandage that contains a nitrocellulose insert on which specific antibodies have been printed for capturing proteins directly from the skin surface. TAP has shown efficacy in detecting soluble skin surface biomarkers on healthy, inflamed, and irritated skin without discomfort (Orro et al., 2014 [17]; Falcone et al., 2017 [18]; Schaap et al., 2021 [19], Schaap et al., 2022 [20]; Orro et al., 2023 [21]). 

The aim of the present pilot study was to assess whether the expression of antimicrobial hBD-1 and hBD-2 and the inflammatory signaling molecules IL-1α, CXCL-1/2, and CXCL-8, known to be involved in psoriasis pathogenesis, are associated with disease severity and whether the expression patterns of these proteins on the skin surface can be used to measure pharmacodynamic effects of biological therapy.

## 2. Results

### 2.1. Human Beta-Defensins and Inflammatory Chemokines CXCL-1/2 and CXCL-8 Are Highly Expressed on the Surface of Psoriatic Lesions

The skin-surface expression of hBD-1, hBD-2, IL-1α, CXCL-1/2, and CXCL-8 was measured on lesional skin (and non-lesional skin of PV patients (N = 37)) using FibroTx Transdermal Analysis Patches. Comparing the levels of skin surface hBD-1 and hBD-2 captured from psoriasis patients’ lesional and non-lesional skin revealed notably higher levels of hBD-1 (0.20 vs. 0.11 ng/mL) and hBD-2 (4.48 vs. 1.48 ng/mL) psoriasis lesions when compared to the non-lesional skin of the same patients (*p* < 0.05 and *p* < 0.0001, respectively).

Similarly, proinflammatory chemokines CXCL-1/2 and CXCL-8 were detected in higher amounts on PV patients’ lesional skin when compared to non-lesional skin (0.03 ng/mL vs. ND, *p* < 0.0001, and 0.25 vs 0.01 ng/mL, *p* < 0.0001, respectively). Contrarily, decreased levels of IL-1α were documented on the lesional skin compared to levels detected on patients’ non-lesional skin (1.94 ng/mL vs 0.21 ng/mL, *p* < 0.0001; see Figure 1).

### 2.2. Psoriasis Area and Severity Index (PASI) and Local Scores of Induration, Desquamation, and Erythema Are Associated with the Levels of Biomarkers Detected on Psoriatic Lesions 

To assess the relationship between skin surface measurements of psoriatic skin and severity of psoriasis, we analyzed the correlation between the values of hBD-1, hBD-2, IL-1α, CXCL-1/2, and CXCL-8 measurements from psoriatic skin lesions against the PASI scores and local scores of induration, desquamation, and erythema at the area of TAP measurements, as assessed by a dermatologist. The correlation between PASI and skin surface biomarkers captured by TAP was explored using a robust Spearman’s rank-correlation analysis (Figure 2). A significant positive correlation of PASI was found with hBD-2 (*Sr* = 0.5349, *p* < 0.0001), and a moderate positive correlation was found with CXCL-1/2 (*Sr* = 0.3821, *p* < 0.0001). A moderate negative correlation was found between PASI and IL-1α (*Sr* = −0.3571, *p* < 0.001).

Correlating the values of hBD-1, hBD-2, IL-1α, CXCL-1/2, and CXCL-8 measurements from psoriatic skin against the values of induration, desquamation, and erythema at the area of skin surface protein measurements revealed a notable positive correlation between the clinical score of induration, as well as for erythema and the levels of hBD-2, CXCL-1/2, and CXCL-8 on psoriatic lesions. A clear negative correlation between the levels of IL-1α captured in the psoriatic lesions and clinical scores of induration, desquamation, and erythema was determined (Table 1).

### 2.3. The Efficacy of Biological Therapy Is Reflected in the Clinical Scores and the Levels of Skin Surface Proteins Detected on Psoriatic Lesions 

Over the treatment period, therapy resulted in notable changes in PASI scores as well as in lesion-specific local clinical scores in psoriasis patients. As a result of biological treatment, PASI scores decreased by an average of 91.36% when compared to baseline (Figure 3, panel A). The decrease in local erythema, induration, and desquamation in response to therapy had a similar response pattern as the PASI scores when compared to baseline (Figure 3, Panel A–D). It is important to note that there were no significant differences in PASI scores or local inflammation scores between responders and non-responders at baseline (*p* > 0.05).

By the end of the treatment monitoring period, 30 of the 37 enrolled patients (81.08%) achieved at least a 75-point reduction in PASI scores (PASI75), and 7 patients (18.91%) did not respond to the therapy (PASI reduction < 75 points). Of the 30 psoriasis patients who achieved PASI75, 20 achieved PASI90 or higher (super responders). As a result of the biological therapy, the clinical scores dropped by an average of over 90% in the patient groups of PASI-75 and PASI-90 (Figure 3 panel A–D). The PASI scores of non-responders reduced by an average of 59.74%; however, no significant reduction in induration, desquamation, nor erythema was observed for the final sampling point when compared to the levels documented at baseline (Figure 3, panel B–D). 

To determine whether the efficacy of the response to biologics therapy is manifesting only in clinical scores or is reflected in skin surface biomarker levels as well, the skin surface protein analysis was performed by first analyzing the biomarker data of the whole cohort (N = 37), responders (PASI75) and non-responders (PASI < 75). A notable decrease in response to biologics therapy in the levels of hBD-2 (4.52 vs. 1.41 ng/mL), CXCL-1/2 (0.035 vs. 0.009 ng/mL), and CXCL-8 (0.25 vs. 0.08 ng/mL) was observed in the overall cohort when compared to levels detected at baseline (Figure 4, panels A,C,D). A continuous increase in IL-1α levels was detected in psoriatic lesions during the treatment period (0.22 ng/mL to 1.38 ng/mL), leading to a 5-fold increase compared to the IL-1α levels detected at baseline (Figure 4, panel B).

The levels of antimicrobial hBD-1 detected on the lesional skin of PV patients did not present any firm response trend over the therapy (Appendix A).

The levels of skin surface hBD-2, CXCL-1/2, and CXCL-8 captured in the lesional skin of responders decreased notably in response to the biological therapy (Figure 4, panels A,C,D). The levels of skin surface hBD-2 reduced over the therapy by an average of 2.44-fold compared to the levels detected before therapy. The levels of CXCL-1/2 and CXCL-8 presented a rapid decrease, reducing from 0.04 ng/mL to 0.002 ng/mL and 0.27 ng/mL to 0.083 ng/mL, respectively. Contrarily, the levels of IL-1α detected in lesional skin increased over the therapy from an average of 0.14 ng/mL to 1.58 ng/mL, presenting a notable upregulation of this proinflammatory cytokine. The biomarker response pattern of the super responders was very similar to the responders group biomarker pattern and kinetics (Figure 4, panel A–D).

Analysis of the biomarker levels of the non-responders group revealed similar response patterns up to the therapy maintenance phase, except the levels of hBD-2 and IL-1 α (Figure 4, Panel A–D). The levels of proinflammatory chemokines CXCL-1/2 and CXCL-8 reduced shortly after treatment initiation (induction phase) and presented a notable and continuous increase in their levels in the therapy maintenance phase, reaching from 0.003 ng/mL (T1) to 0.035 ng/mL (T5) and from 0.076 ng/mL (T1) to 0.10 (T5) ng/mL, respectively. The levels of IL-1α presented a minor increase (0.57 ng/mL (T0) vs 1.13 ng/mL (T3)) until the therapy maintenance phase, followed by a steady decrease (Figure 4, panel B). The levels of human antimicrobial peptide hBD-2 presented a fast response to the therapy induction phase, and the subsequent reversal in concentrations was faster than in the case of CXCL-1/2 or CXCL-8 (Figure 4, panel A,C,D). 

### 2.4. The Biomarker Response Kinetics to Biological Therapy Is Influenced by Applied Therapeutics

In the current study, the response of clinical scores and skin surface biomarkers to biologics targeting TNF-α, IL-12/IL-23, and IL-17A were monitored. The average PASI score, as well as the average induration, desquamation, and erythema scores, of all three therapeutic target groups presented a gradual decrease over the course of therapy (Appendix A).

Analyzing the response of the biomarkers of skin lesions according to the therapeutic targets revealed different response kinetics. The average amount and response kinetics of hBD-2 in response to anti-TNF-α- and anti-IL-12/23-targeted therapy was very similar (Figure 5, Panel A), presenting slight reduction over the induction phase and remaining at the same level for the maintenance phase until the end of the 32-week or 52-week period, respectively. Contrarily, levels of hBD-2 influenced by anti-IL-17A therapy presented an abrupt decrease in the induction phase in response to therapy (Figure 5, panel A, Figure 6, panel A).

Similarly, the levels of IL-1α in response to targeted IL-17A therapy present different kinetics when compared to the anti-TNF-α- and anti-IL12/23-targeted therapy (Figure 5, panel B, Figure 6, panel B): a fast increase in the average level of lesional IL-1α was observed only in response to anti-IL-17A-targeted therapy initiation (induction phase), whereas a continuous increase in the biomarker levels by anti-TNF-α and anti-IL12/23 therapy was noted over the course of treatment (Figure 5, panel B).

The average levels of CXCL-1/2 and CXCL-8 captured on the lesional skin of psoriasis patients subjected to anti-TNF-α-targeted therapy present a continuous decrease in the induction phase and an increase in the maintenance phase (Figure 5 and Figure 6, panels C and D, respectively). However, the levels of CXCL-1/2 and CXCL-8 in response to anti-IL-17A did not present any such increase in the maintenance phase (Figure 5, panels C and D). There was a small increase in the levels of CXCL-8 during the maintenance phase of the anti-IL-12/23 therapy (Figure 5, panel D). No such pattern was noted for the PASI nor the clinical scores in the maintenance phase (Appendix A).

## 3. Discussion

The main goals of the current study were to assess whether expression patterns of skin surface hBD-1, hBD-2, IL-1α, CXCL-1/2, and CXC-8, examples of proteins known to be involved in psoriasis pathology, are associated with disease severity and whether expression patterns of these skin surface proteins can be used to measure pharmacodynamic effects of biological therapy.

The evaluation of the efficacy of the psoriasis therapy over the treatment course is primarily performed visually by clinical evaluation. The PASI score covering the severity of local inflammation scores of the skin desquamation, erythema, and induration of inflamed lesions is assessed (Carlin and Feldman, 2004 [22]; Daugaard and Iversen, 2022 [23]), allowing for the monitoring of changes in affected skin areas over time, which may either reflect progression, relapse, or improvement of the disease. However, visual assessment also has its limitations in monitoring skin inflammation since it does not consider the underlying inflammation regulated by various cell signaling processes. Thus, visual assessment of the skin surface can reflect the processes and dynamics in the deeper layers of the skin with pronounced delay. TAP was chosen for measuring skin surface proteins since TAP is a non-invasive sampling technology that does not affect the skin, i.e., protein measurements are not biased by skin responding to the measurement method and do not interfere with biological processes in and on the skin (Falcone et al., 2017 [18]).

The biomarker panel containing hBD-1, hBD-2, IL-1α, CXCL-1/2, and CXCL-8 was chosen for this study because these proteins have shown reliable quantitative and qualitative measurements from the skin surface using TAP (Orro et al., 2014 [17]; Schaap et al., 2021 [19]; Orro et al., 2023 [21]) and based on their role in inflammation of psoriasis reported in the literature (Jansen et al., 2009 [13]; Baliwag et al., 2015 [14]; Chularojanamontri et al., 2019 [16]; Iznardo and Puig 2022 [24]). IL-1α is a proinflammatory cytokine produced by various cell types (including epithelial and immune cells) that has proven to have a significant role in skin homeostasis as well as in inflammation, including in chronic inflammatory skin disorders such as psoriasis. IL-1α can interact with other crucial cytokines, such as TNF-α and IL-17, for example, and activate several inflammatory pathways, including the NF-κB pathway, initiating the production of other proinflammatory cytokines and chemokines, which further contribute to the inflammatory response (Shi et al., 2017 [25]; Villarino et al., 2017 [26]). Chemokines CXCL1/2 and CXCL-8 were selected due to their inflammatory nature and, therefore, their involvement in the pathogenesis of psoriasis. They recruit immune cells like monocytes and T-cells into the inflammation site and neutrophils into the epidermis (Palomino and Martin, 2015 [27]; Zdanowska et al., 2021 [28]). Human beta-defensins 1 and -2 were chosen due to their antimicrobial properties of preventing secondary infections in inflammatory lesions (Chieosilapatham et al., 2019 [29]). Moreover, hBD-2 has been shown to be a serum biomarker for psoriasis activity reaching biologically relevant concentrations in lesional skin (Jansen et al., 2009 [13]; Tingting et al., 2019 [30]).

Biomarker measurements at the baseline of this study revealed distinguishable skin surface protein expression patterns between the PV patients’ non-lesional and lesional skin. Higher levels of antimicrobial hBD-1, hBD-2 and chemokines CXCL-1/2 and CXCL-8 were detected on patients’ psoriatic lesions when compared to the non-lesional skin of the same patients, whereas the levels of proinflammatory IL-1α were higher on non-lesional patients (Figure 1). These findings measured on the skin surface coincide with previously reported protein expression patterns measured by invasive technologies like skin biopsies, immune-histochemistry, and mRNA analysis (Kolbinger et al., 2016 [31]; Hulshof et al., 2019 [32]; Tarentini et al., 2021 [33]), as well as non-invasive measurements (Orro et al., 2014 [17]; Robke et al., 2021 [34]; Schaap et al., 2021 [19]; Orro et al., 2023 [21]). Thus, the skin surface measurements of these proteins reflect the skin status and its regulatory processes.

We monitored patients during biological therapy to determine whether measurements of skin surface hBD-1, hBD-2, IL-1α, CXCL-1/2, and CXCL-8 have the potential to reflect disease severity. Firm association between the expression patterns of hBD-2, IL-1 α, and CXCL-1/2 was established between the PASI score (Figure 2) of the analyzed PV patents referring to these analyzed skin surface proteins, which are directly related to PASI and inflammation. Further, correlation with clinical assessment of erythema, induration, desquamation, and the expression levels of hBD-2, IL-1α, CXCL-1/2, and CXCL-8 of the same lesion were observed (Table 1). Thus, PV patients responding to the biologics therapy presented a consistent and significant decrease in the PASI and clinical scores of local inflammations (erythema, induration, and desquamation) and the levels of skin-surface-captured hBD-2, and chemokines CXCL-1/2 and CXCL-8 followed the same pattern (Figure 3). A firm positive correlation between PASI and detected skin surface hBD-2 levels on lesional skin aligns with previous findings describing serum-detected systemic hBD-2 levels correlating with psoriasis severity (Jansen et al., 2009 [13], Kolbinger et al., 2016 [31]). Thus, it appears that skin surface hBD-2 has the potential to be a useful surrogate marker for disease activity in psoriasis and that the skin surface can reflect the underlying inflammation.

However, no such decrease was noted for lesional IL-1α, the levels of which presented a gradual increase over the therapy. This kind of free IL-1α accumulation could be caused/explained due to the healing of the damaged skin barrier and the importance of this cytokine in the regulation of innate immune defense mechanisms (Boraschi and Tagliabue, 2013 [35]), angiogenesis, and by the constant production of local keratinocytes. Pre-formed IL-1α stored in the deeper layers of the skin is emerging and is constantly produced by activated keratinocytes in the epidermis during the regeneration processes. Further, we have previously demonstrated that the skin surface anti-inflammatory IL-1RA, synthesized in counter to inflammation, reduced significantly in response to therapy (Orro et al., 2023 [21]). This hypothesis seems to be supported by the measurement of IL-1α levels of the same psoriasis patients’ non-lesional skin at the baseline of the healthy skin of the same PV patients.

Interestingly, although skin surface hBD-1 was detected at higher levels on psoriatic lesions than on non-lesional skin of the patients like hBD-2, no significant effect of the biologics therapy was noted for this skin surface antimicrobial peptide expression level. Although there is evidence of the reduction in lesional hBD-1 in response to short-term therapy in inflamed atopic dermatitis and psoriasis lesions (Robke et al., 2021 [34]; Orro et al., 2023 [21]), we could not confirm such effect over long-term biologics therapy. We hypothesize that this discrepancy between hBD-1 levels in psoriasis and atopic dermatitis could be related, on the one hand, to the known differences in infection rate between these two diseases and, on the other hand, to the applied therapy approaches.

The analyzed skin surface proteins fall into distinguished groups with different treatment response kinetics depending on applied therapeutics. The levels of skin surface hBD-2 and IL-1α present faster response kinetics regarding IL-17A-targeted therapy when compared with the response kinetics of IL-12/23- and TNF-alpha-neutralizing antibody therapy (Figure 5, panel A,B) whereas such observations could not be made for the detected skin surface levels of CXCL-1/2 and CXCL-8 of the same PV patients (Figure 5, panel C,D). The response kinetics of skin surface CXCL-8 presented faster downregulation to anti-IL-17A- and anti-IL-12/23-targeted therapy and levels of CXCL-1/2 to anti-IL-12/23-targeted therapy (Figure 5, panel C,D).

This discrepancy between biomarker regulation levels could be explained, on one hand, by previous findings in the literature describing studies documenting the higher efficacy of IL-17 and IL-12/IL-23 inhibitors for the treatment of psoriasis compared to other biologics (Lebwohl et al. 2015 [36], Langley et al. 2014 [37], Bergen et al. 2020 [38], ong et al., 2022 [39]) and, on the other hand, by the origin of cell type in skin producing majority of the analyzed signaling molecules. Although CXCL-1/2 and CXCL-8 are produced by epidermal keratinocytes, these proinflammatory signaling molecules are predominantly synthesized by the immune cells infiltrated to the inflamed lesion and immune cells origin deeper in the skin, which take time to build up on the skin surface compared to hBD-2 and IL-1α, which are produced predominantly by epidermal keratinocytes.

Interestingly, we noticed some increase in the levels of CXCL-1/2 and CXCL-8 captured on psoriasis lesions of the patients subjected to anti-TNFα therapy, whereas there was no such increase noted on the PASI nor local scores of patients. Further, no skin surface CXCL-1/2 and CXCL-8 increase was observed for the IL-17A- or IL-12/23-targeted therapy group. These occurrences between treatment groups can be explained due to the anti-drug antibody (ADA) formation, where antibodies over the TNF-α-targeted treatment course against Infliximab and Adalimumab have been formed and decreased the clinical response.

Such observation indicates that skin surface proteins reflect the underlying skin inflammation earlier compared to visual clinical evaluation and could therefore contribute to monitoring the disease status and “molecular footprint” for skin diseases, including psoriasis.

In conclusion, we could detect skin surface IL-1α at lower levels and hBD-1, hBD-2, CXCL-1/2, and CXCL-8 at notably higher levels on the psoriatic lesions when compared to the non-lesional skin of the same PV patients, which supports the hypothesis that non-invasive biomarker sampling along with skin surface protein analysis can be an applicable method for skin status analysis. Moreover, correlating skin surface measurements of analyzed hBD-2, CXCL-1/2, and CXCL-8 on the skin from psoriasis patients with clinical assessments of psoriasis severity indicates that these protein measurements may have potential as biomarkers for monitoring disease severity over the treatment course and could describe the underlying molecular footprint of psoriasis prior to visual clinical assessment. Nevertheless, we point out that the patient cohort of the current study was limited, and a substantially larger study with a larger cohort of patients is needed for firm conclusions and to validate the measurements of IL-1α, IL-1RA, CXCL-1/2, and CXCL-8 on the skin of psoriasis patients for diagnostic and therapeutic biomarker applications. Whether the conclusions about the applicability of the TAP method and analyzed biomarkers can be applied to other skin diseases, skin surface biomarkers, and/or treatment methods is not known and will need further research.

## 4. Materials and Methods

### 4.1. Study Participants 

This study was an explorative, observational, non-invasive study performed at the Dermatology Clinic of Tartu University Hospital in Estonia between 2016 and 2020 under the approval of the Research Ethics Committee of the National Institute for Health Development (Application no 1594, Decision No. 1420). A total of 37 adult patients with moderate to severe PV visiting a dermatologist at the Dermatology Clinic of Tartu University Hospital were included in this study. The patients included in this study had not received any systemic form of medical treatment or any kind of phototherapy for at least 4 weeks prior to this study and had not received any topical form of medical treatment for at least 2 weeks prior to this study. Pregnant or breastfeeding women and volunteers with a history of other skin diseases were excluded from participation. Detailed information regarding the participants is shown in Appendix A.

### 4.2. Transdermal Analysis Patch Measurements and Skin Disease Severity Scores 

In order to determine the differences in skin surface biomarkers, FibroTx transdermal analysis patch (TAP, see Appendix A) antibody micro-arrays (FibroTx LLc., Tallinn, Estonia) coated with anti-hBD-1 (PeproTech EC Ltd., London, UK), anti-hBD-2 (PeproTech EC Ltd., London, UK), interleukin one alpha (IL-1α) (PeproTech, London, UK), anti-chemokine (C-X-C motif) ligand 1 and 2 (CXCL-1/2) (PeproTech, London, UK), and anti-chemokine (C-X-C motif) ligand 8 (CXCL-8) (PeproTech EC Ltd., London, UK) were incubated for 20 min on the non-lesional and lesional skin of psoriasis patients. Captured skin surface proteins were visualized using spot-enzyme linked immunosorbent assay (ELISA), as previously described (Orro et al., 2014 [17]). 

All FibroTx TAP and clinical measurements were performed right before a new treatment dose. Measurements were performed on the same position on the skin at each time point. In parallel, visual scores of induration, desquamation, and erythema (0–4 scale) for local inflammation sore were performed at the exact location of the TAP measurements. In addition, the PASI score (the range of PASI scores is 0–72) was assessed and documented by the dermatologist.

### 4.3. Biological Therapy

In the current study, the effect of anti-tumor necrosis factor (TNF)-α, anti-IL-17A, and anti-IL-12/IL-23-targeted therapeutics on clinical scores (PASI, erythema, desquamation, induration) and on the levels of skin surface hBD-1, hBD-2, IL-1α, CXCL-1/2, and CXCL-8 were monitored. Anti-TNF therapy (adalimumab and infliximab), anti-IL-17A (secukinumab), and anti-IL-12/23 therapy (ustekinumab) were administered by using the guidelines of the standard psoriasis treatment regimen (Nast et al., 2022 [40]).

Patients were followed during their treatment period, and the analysis points for clinical scores and skin surface biomarker analysis of each biological therapy were as follows:Anti-TNF-α-targeted therapy (infliximab (N = 8), and adalimumab (N = 15); Baseline (T0), Week 2 (T1), Week 4 (T2), Week 12 (T3), Week 24 (T4), Week 32 (T5);Anti-17A-targeted therapy (secukinumab): Baseline (T0), Week 2 (T1), Week 4 (T2), Week 16 (T3), Week 32 (T5);Anti-IL-12/23-targeted therapy (ustekinumab): Baseline (T0), Week 4 (T2), Week 16 (T3), Week 28 (T4), Week 40 (T5), Week 52 (T6).

Sampling points from baseline (T0) to week 16 (T1–T3) are the induction phase and weeks 24 to 52 (T4–T6) are the maintenance phase of the biologics therapy.

The efficacy of biological therapy was determined based on PASI scores achieved at the endpoint of the treatment: patients who achieved PASI below 75 (non-responders) and patients who achieved PASI-75 or higher (responders to therapy). Further, the biomarker levels of the subjects who achieved PASI-90 (super responders) were analyzed independently as well. 

### 4.4. Statistical Analyses 

All statistical tests were performed using the statistics program GraphPad Prism 9 (version 9.5.1 (528) for macOS). For statistical analysis, the normality of the data was tested with the Shapiro–Wilk test. Statistical significance for related groups analysis was determined by using matched paired Wilcoxon signed-rank test. The Mann–Whitney non-parametrical test was applied for two unrelated groups. For correlation analysis, non-parametrical Spearman’s Rank correlation analysis was performed, and statistical significances were verified with probability value (p-value). Correlation coefficients were interpreted based on Cohen (Cohen, 1988 [41]) to indicate weak (0.1), moderate (0.3), and strong (0.5) correlations. The level of statistical significance was set at 5% (*p* < 0.05). 

## Figures and Tables

**Figure 1 ijms-24-16248-f001:**
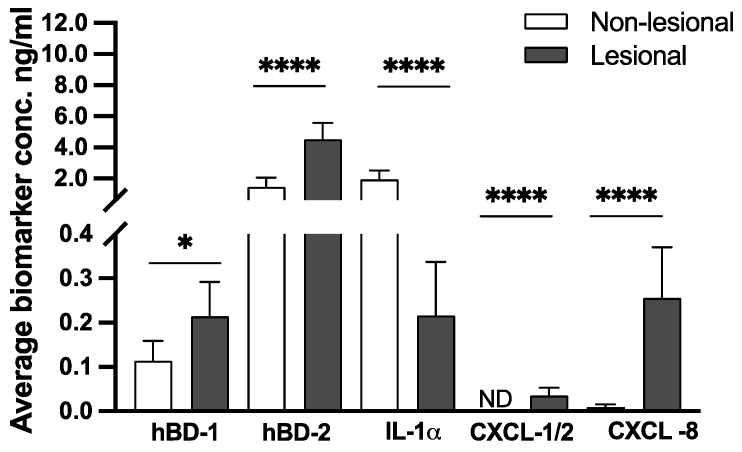
Skin surface average biomarker concentration measured at baseline with Transdermal Analysis Patches on lesional and non-lesional skin of psoriasis patients (N = 37). The average biomarker concentrations of hBD-1, hBD-2, IL-1α, CXCL-1/2, and CXCL-8 detected from non-lesional skin (white bars) and lesional skin (dark grey bars) have been blotted. *Y*-axis: Average concentration of analyzed biomarker at baseline on the skin surface in ng/mL. *X*-axis: sampled biomarker. Error bars on graphs present the 95% confidence intervals for the average of combined measurements in the participants. A statistically significant difference (paired sample *t*-test) between biomarker expression on lesional and non-lesional skin is shown with asterisks: * *p* < 0.05, **** *p* < 0.0001). ND—not detected.

**Figure 2 ijms-24-16248-f002:**
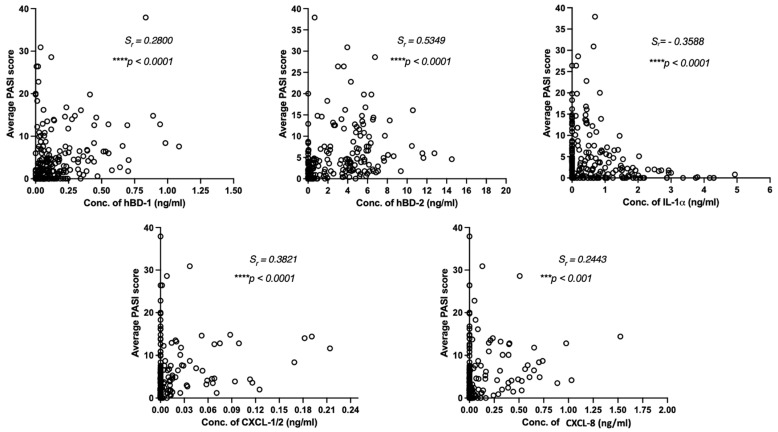
Correlations between Transdermal Analysis Patch measurements of hBD-1, hBD-2, IL-1α, CXCL-1/2, and CXCL-8 on lesional skin and Psoriasis Activity and Severity (PASI) scores over the course of biologics therapy. Spearman’s correlation coefficient (*S_r_*) was calculated between PASI and for each skin surface protein detected on lesional skin of psoriasis patients (N = 37) over the course of biological treatment. Data collected at the baseline and all following analysis points (T1–T6) were combined for Spearman rank correlation analysis (*Sr*). Statistical significance was verified with probability value (*p*-value), *** *p* < 0.001, **** *p* < 0.0001).

**Figure 3 ijms-24-16248-f003:**
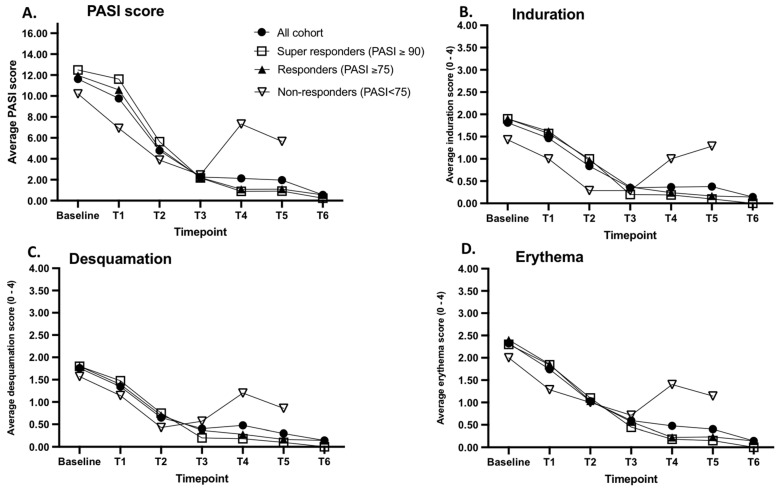
Changes in the psoriasis area severity index (PASI) and local scores of induration, desquamation, and erythema induced by biological therapy. The PASI (panel **A**), and local scores of induration (panel **B**), desquamation (panel **C**), and erythema (panel **D**) were documented before the treatment initiation (baseline) and at each time point at the same lesion sites. Each plotted line represents a combined average measurement of a clinical score of all analyzed psoriasis patients (black circles, N = 37), super responders (white squares, N = 20), responders (black triangles, N = 30), and non-responders (white triangles, N = 7). *Y*-axis: Average clinical score of analyzed patients on the lesional skin, *X*-axis: sampling time point.

**Figure 4 ijms-24-16248-f004:**
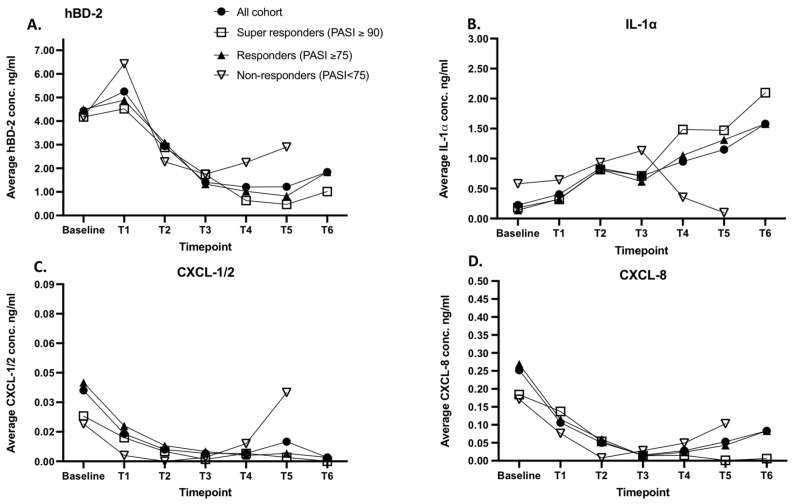
Measurements of skin surface hBD-2, IL-1α, CXCL-1/2, and CXCL-8 on lesional skin of psoriasis patients using FibroTx TAP. The biomarker concentrations of hBD-2 (panel **A**), IL-1α, (panel **B**) CXCL-1/2 (panel **C**), and CXCL-8 (panel **D**) were detected from the skin surface of psoriatic lesions in 37 psoriasis patients over the course of biologic therapy. The measurements of skin surface hBD-2, IL-1α, CXCL-1/2, and CXCL-8 were documented before the treatment initiation (baseline) and before each of the following therapy sessions (T1–T6) at the same lesions of all enrolled psoriasis patients (black circles, N = 37), super responders (white squares, PASI ≥ 90, N = 20), responders (black triangles, PASI ≥ 75, N = 30), and non-responders (white triangles, PASI < 75, N = 7). *Y*-axis: Combined average concentration of analyzed biomarker on psoriasis lesions in ng/mL. *X*-axis: sampling time point.

**Figure 5 ijms-24-16248-f005:**
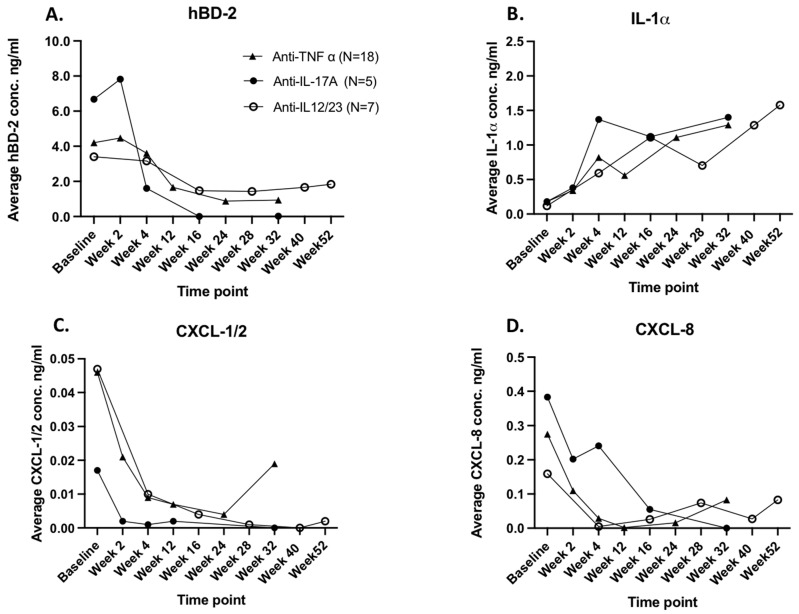
Measurements of hBD-2, IL-1α, CXCL-1/2, and CXCL-8 over the course of biologics treatment on lesional skin of psoriasis patients according to therapy target, using FibroTx TAP. The average skin surface concentrations of hBD-2 (panel **A**), IL-1α (panel **B**), CXCL-1/2 (panel **C**), and CXCL-8 (panel **D**) in response to the anti-TNF-α-targeted therapy (black triangles, N = 18), anti-IL17A-targeted therapy (black circles, N = 5), and anti-IL-12/23-targeted therapy (white circles, N = 7) were analyzed from the patients’ psoriasis lesions. *Y*-axis: Combined average concentration of analyzed biomarker on psoriasis lesions in ng/mL. *X*-axis: sampling time point.

**Figure 6 ijms-24-16248-f006:**
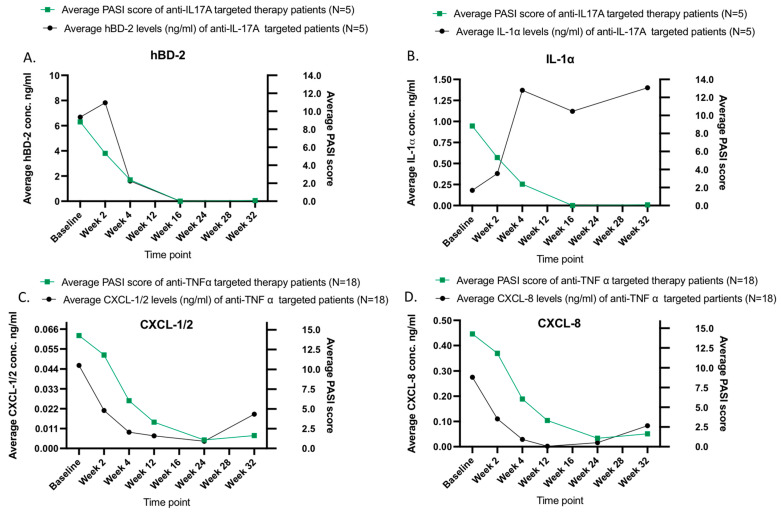
Measurements of skin surface hBD-2, IL-1α, CXCL-1/2, and CXCL-8 on psoriasis lesions, and PASI scores in response to therapy according to the biologic targets using FibroTx TAP. The average biomarker concentration of hBD-2 (panel **A**) and IL-1α (panel **B**) captured from the lesional skin of psoriasis patients (N = 8) over the course of IL-17A-targeted therapy combined with the average PASI score (panels **A**,**B**, green squares) of the same patients. The average biomarker measurements of CXCL-1/2 (panel **C**, black circles) and CXCL-8 (panel **D**, black circles) captured from the lesional skin of psoriasis patients (N = 18) over the TNF-α-targeted therapy combined with the average PASI scores (panels **C**,**D**, green squares) of the same patients. *Y*-axis: Combined average concentration of analyzed biomarker on psoriasis lesions in ng/mL. *X*-axis: sampling time point.

**Table 1 ijms-24-16248-t001:** Correlations between FibroTx TAP measurements of hBD-1, hBD-2, IL-1α, CXCL-1/2, and CXCL-8 on lesional skin and the local scores of induration, desquamation, and erythema in psoriasis patients. Data collected at the baseline and at all following analysis points (T1–T6) were combined for Spearman rank correlation analysis (*Sr*). Statistical significance was verified with probability value (*p* < 0.05, ** *p* < 0.01, *** *p* < 0.001, **** *p* < 0.0001).

Biomarker	Induration		Desquamation		Erythema	
	*Sr*	*p*	*Sr*	*p*	*Sr*	*p*
hBD-1	0.0004	*p* *** = 0.0004	0.217	*p* ** = 0.0015	0.2722	*p* **** < 0.0001
hBD-2	0.4863	*p* **** < 0.0001	0.4821	*p* **** < 0.0001	0.5165	*p* **** < 0.0001
IL-1a	−0.4589	*p* **** < 0.0001	−0.4417	*p* **** < 0.0001	−0.4401	*p* **** < 0.0001
CXCL-1/2	0.4854	*p* **** < 0.0001	0.3679	*p* **** < 0.0001	0.4591	*p* **** < 0.0001
CXCL-8	0.3401	*p* **** < 0.0001	0.2564	*p* *** = 0.0002	0.3786	*p* **** < 0.0001

## Data Availability

Data are contained within the article and Appendix A.

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
