# Peer review of "Non-Invasive Assessment of Skin Surface Proteins of Psoriasis Vulgaris Patients in Response to Biological Therapy"

_ijms, 2023, doi:10.3390/ijms242216248_

Round 1
Reviewer 1 Report
Comments and Suggestions for Authors
In this article, Orro et al assessed whether expression patterns of skin surface hBD-1, hBD-2, IL-1α, CXCL-1, and CXCL-8 are associated with the disease severity and whether expression patterns of these proteins on the skin surface can be used to measure pharmacodynamic effects of biological therapy. Significantly higher levels of hBD-1, hBD-2, CXCL-1/2, and CXCL-8 were detected on lesional skin compared to the non-lesional skin of the PV patients. In contrast, lower levels of IL-1α were found in lesional skin compared to non-lesional skin. The biomarker expression levels correlate with the disease severity. Changes in the expression levels of skin surface biomarkers during biological therapy correlate with treatment response. Biomarker expression patterns in response to treatment differed somewhat between treatment subtypes. In the case of anti-TNF-α therapy, an increase after a steady decrease in the expression levels of CXCL-1/2 and CXCL-8 occurred before the change in clinical scores. Moreover, response kinetics of skin surface proteins differs between applied therapy. hBD2 expression responds quickly to anti-IL-17A therapy, CXCL-1/2 to anti-IL-12/23, and levels of CXCL-8 are rapidly down-regulated by IL-17A and IL-12/23 therapy. They concluded that the skin surface hBD-2, IL-1α, CXCL-1/2, and CXCL-8 are markers for the psoriasis severity. Skin surface proteins CXCL-1/2 and CXCL-8 may have value as therapeutic biomarkers thus confirming that measuring the ‘molecular root’ of inflammation appears to have value scoring disease severity on its own. It is very interesting that cytokines can be measured by tape stripping and that this may predict things such as the skin lesions that may be exacerbated. I have several questions as follows.
major concerns)
1) Is it possible to analyze which cytokines are more effective in improving skin lesions and predicting future flare-ups, using multivariate analysis? If possible, please analyze the data, so that it is not just an impression, but data backed up by analysis, which will make the data more reliable.
2) In each graph, there are only dots and lines drawn for the values. While the average trend is easy to understand, the distribution of each value is difficult to understand. I would appreciate it if you could use an easy-to-understand presentation method, such as adding error bars or indicating 95% CIs. It would also be helpful to put a mark where there is significance.
minor concerns)
1) In line 166, "n psoriasis lesions when compared to the non-lesional skin of the same patients (p < 0.05 and p < 0.0001..." Is the "n" some kind of typo? Or is there some letter in there? Please correct appropriately.
2) In Figure 6A and 6B, "Average hBD-2 levels (ng/ml) anti-IL-17A targeted patients (N=5)" is missing "of". Please correct appropriately.
Comments on the Quality of English LanguageThere are several typos and notations that need to be corrected.
Reviewer 2 Report
Comments and Suggestions for Authors
Thank you for the opportunity to review this manuscript by Drs Orro and colleagues. These studies used a novel non invasive patch (FibroTx TAP) to assess levels of several biologic agents in psoriatic patients to gauge their response to biologic therapies. The studies are highly significant in that a non invasive set of markers could be a useful adjunct along with clinical parameters in assessing response to therapy. The results were that they could detect skin surface hBD-1, hBD-2, IL-1α, CXCL-1/2, and CXCL-8 levels. All proteins except IL-1a were increased in psoriasis lesions and they normalized in response to therapy. The markers chosen are appropriate.
Though the numbers of subjects in this study are relatively small, the current work has considerable value. The methodology and analysis appear appropriate. I have only a few minor concerns with this study.
Minor issues.
Figure 5 a. the authors note that hBD2 levels had an “abrupt decrease” in response to IL17A inhibition. Yet, the numbers of subjects (5) and that the hBD2 levels at baseline were so much higher than those for other therapies—the authors need to mention these issues as this is not overly convincing.
Lines 444-445—The authors seem to be calling attention to the increased levels of CXCL1/2 and CXCL-8 at week 32 treated with TNF inhibitors. However, it is not clear that the data in Figures 6 c,d are significant (especially for CXCL-8). This mechanism was proposed to be Anti-Drug Antibodies (ADA), yet were ADA levels measured?
Lines 452-453—IL-1 was not found at higher levels, but lower levels.
Reviewer 3 Report
Comments and Suggestions for Authors
Ms-items-2691541 can be accepted with minor revisions. The information in the introductory part is well articulated and supported by a decent bibliography, in fact, the recent bibliographic sources, from the last 5 years, are 11 out of 44.
- the methods described by the authors are appropriate and clearly described.
- we recommend using a table to collect the indications present in 2.3 Biological Therapy, from lines 136 to 141.
- how is the high conc explained? of hBD1-2 and IL-1a in the Non-lesional PV patients presented in Fig.1?
-In anti-TNFα, anti-IL17, and anti-IL12/IL23 therapy, what concentration of the drug is used, and how many applications per day?
- Furthermore, in agreement with the authors, the conclusions of this MS should be re-evaluated considering a greater number of patients than that presented in this study
-The text must be rechecked for the presence of some form and typing errors
Comments on the Quality of English LanguageThe text must be rechecked for the presence of some form
